# Watershed Horizontal Ecological Compensation Policy and Green Ecological City Development: Spatial and Mechanism Assessment

**DOI:** 10.3390/ijerph20032679

**Published:** 2023-02-02

**Authors:** Xinwen Lin, Angathevar Baskaran, Yajie Zhang

**Affiliations:** 1Department of Development Studies, Faculty of Business and Economics, Universiti Malaya, Kuala Lumpur 50603, Malaysia; 2Department of Development Studies, Faculty of Business and Economics & UM North–South Research Centre, Universiti Malaya, Kuala Lumpur 50603, Malaysia; 3SARChI (Innovation Studies), Tshwane University of Technology, Pretoria 0183, South Africa; 4Department of Economics and Management, Sanming Medical and Polytechnic Vocational College, Sanming 365000, China

**Keywords:** watershed horizontal ecological compensation, DID, mechanism assessment, spatial analysis

## Abstract

Green ecological development has become an inevitable choice to achieve sustainable urban development and carbon neutrality. This paper evaluates the level of green ecological city development in the Xin’an watershed as measured by green total factor productivity (*GTFP*), analyzes the direct and spatial effects of the Watershed Horizontal Ecological Compensation policy on *GTFP*, and further examines the moderating effect of the Research and Development (*R&D*) incentives, industrial structure, and income gap. This paper conducts difference-in-differences (DID) and spatial regression analysis on 27 cities from 2007 to 2019. The results show that *GTFP* progresses to varying degrees across cities over time, especially in the pilot cities. Crucially, the Watershed Horizontal Ecological Compensation policy significantly improved *GTFP*, although the effect was slight. Interestingly, the increase in *GTFP* in pilot cities that implemented the policy spatially suppressed the increase in *GTFP* in cities that did not implement the policy. Our evidence also shows that the positive effect of the policy is higher in regions with higher *R&D* incentives and industrial structure upgrading, which indicates that *R&D* incentives and industrial upgrading are crucial. In comparison, the income gap has not made the expected negative adjustment effect under the Chinese government’s poverty alleviation policy. However, the positive policy effect is heterogeneous in the downstream and upstream pilot cities. The “forcing effect” of the policy on the downstream cities is more favorable than the “compensating effect” on the upstream cities. Therefore, policymakers should pay more attention to ensuring the effectiveness of the Watershed Horizontal Ecological Compensation policy in enhancing *GTFP* as a long-term strategy to guarantee the sustainability of green ecological development in Chinese cities.

## 1. Introduction

China sacrificed the environment and consumed resources excessively to achieve rapid economic growth in the early stage of development [1]. During the process of industrialization, a large amount of wastewater flowed into rivers, causing ecological pollution [2,3]. Similarly, urbanization, deforestation and occupation of cultivated land caused a large amount of soil erosion [4], hurting China’s water ecosystem. As China realized the great importance of the concept of green ecology, the Chinese government embarked on the road of ecological governance. In 1979, China promulgated the first environmental law, “Environmental Protection Law of the People’s Republic of China (Trial Implementation)” [5]. In 1982, it formally established a sewage charging system [6]. In 1994, it raised the overall strategy of sustainable development to a national strategy [7]. After that, in 2014, China revised the Environmental Protection Law again [8]. In 2018, it began to collect environmental protection tax [9]. In 2020, it was officially announced that China would strive to achieve carbon peaking by 2030 and carbon neutrality by 2060 [10]. During this process, the Chinese government has paid more and more attention to the ecological governance of water ecosystems. However, there are natural obstacles to the ecological governance of water ecosystems because rivers span different administrative regions geographically, and pollution in upstream basins often causes external costs to downstream basins [11]. Coupled with changes in China’s real estate market structure these factors have led to more difficult green and sustainable urban development [12,13]. Therefore, the ecological governance of rivers requires cross-governmental co-operation. In this context, in 2011, under the leadership of the Ministry of Finance and the Ministry of Environmental Protection of the Chinese central government, Anhui Province and Zhejiang Province signed China’s first cross-provincial watershed horizontal ecological compensation agreement: the Xin’an Watershed Horizontal Ecological Compensation (WHEC). The policy was piloted in the upstream city of Huangshan and the downstream city of Hangzhou (Figure 1). The two provincial governments jointly set up a special compensation fund for the optimization of the industrial layout of the Xin’an Watershed, comprehensive watershed management, water pollution control, and ecological protection. Since the policy was put forward, many scholars have done research on watershed ecological governance [14,15]. However, they have not discussed the heterogeneity of the policy’s effects on ecological governance in upstream and downstream cities, and research on the potential spatial correlation is not sufficient. Based on this, this study uses data covering 27 cities in Zhejiang and Anhui provinces from 2007 to 2019 to analyze the direct and spatial impact of the WHEC policy on green total factor productivity (GTFP) using the difference-in-differences (DID) method. We also carried out a mechanism analysis to verify whether the WHEC policy can be an effective means for China to achieve the goal of green ecological city development and carbon neutrality.

This paper contributes in the following ways. Firstly, in the face of increasingly severe resource and environmental constraints, *GTFP*, which considers unexpected outputs, can better evaluate the city’s green ecological development level, and studying cross-regional co-operation between governments can provide a reference for the Chinese government to achieve green ecological city development and carbon neutrality. Secondly, in terms of methodology, this study adopts the DID method to test the net effect of the policy between pilot cities and non-pilot cities. The WHEC policy was piloted in Huangshan City and Hangzhou City. Therefore, these two cities were selected as the treatment group, and the other 25 cities were used as the control group to evaluate the impact of the WHEC policy and verify the heterogeneity of the “compensating effect” and the “forcing effect” of the policy on the upstream and downstream pilot cities. Likewise, this study performs various tests to ensure the robustness of the estimated results. Thirdly, it examines the spatial effects of *GTFP* by using a mechanism analysis. This study also examines *R&D* incentives, industry structure, and income inequality as channels for improving *GTFP*. Furthermore, this paper helps study the potential spatial spillover effects of *GTFP* to evaluate the policy more comprehensively.

This paper is further organized as follows. Section 2 discusses the literature on the policy context of WHEC and how it affects green ecological city development to form the theoretical basis for this study. Section 3 describes the research methodology, while Section 4 discusses the research results. The last section summarizes the research findings and draws conclusions.

## 2. Literature Review and Conceptual Framework

### 2.1. Ecological Compensation Policy

Ecological compensation aims to protect the ecological environment and to promote harmonious coexistence between man and nature [16,17]. Ecological compensation includes several aspects: first, economic compensation for the protection (restoration) or destruction of the ecological environment [18]; second, internalization of external ecological benefits and costs [19]; third, protection of ecosystems and environments for individuals or regions [20]; and fourth, making protective investments in areas or objects of great ecological value [21]. The practice of ecological compensation in watersheds in China started relatively late, and most of the horizontal compensation is dominated by the government, adopting a two-way compensation model [22]. The upstream and downstream governments of the watershed sign a compensation agreement under the supervision or leadership of the central government. The agreement stipulates that if the upstream implements environmental protection actions and successfully guarantees the water quality of the downstream, the downstream province must pay the corresponding ecological compensation to the upstream province. If the upstream fails to provide water quality that meets the standards, the upstream is responsible for compensating the environmental damage suffered by the downstream province [23]. Since the Xin’an watershed was identified as the first cross-provincial watershed ecological compensation pilot area in 2011, governments have continuously increased environmental protection and promoted collaborative upstream and downstream governance. These measures have preliminarily achieved full coverage of ecological compensation in crucial areas such as forests, wetlands, water flow, cultivated land, the atmosphere, development-prohibited areas, and critical ecological function areas. During the pilot period, the threshold for water quality assessment standards became stricter. The compensation funds of cities in Zhejiang and Anhui increased from CNY 100 million in the first round to CNY 200 million in the second and third rounds. These funds are earmarked for optimization of industrial layout in the Xin’an watershed, comprehensive watershed management, water pollution control, and ecological protection.

Cross-watershed ecological compensation is an essential tool for watershed ecological management. Watershed ecological management is closely related to the maintenance of ecosystem biodiversity and cyclic development [24,25,26]. Additionally, fisheries development [27], agricultural irrigation [28] and domestic water [29] are inseparable from this tool. However, the property rights of the same watershed belong to different administrative regions. Therefore, the success of cross-watershed ecological management requires the co-operation of governments [26]. Meanwhile, the property rights of rivers need to be clarified because the pollution of upstream watersheds is likely to cause external problems to downstream watersheds [30]. Consequently, the essence of cross-watershed ecological governance is to make the upstream and downstream watersheds common property among different governments and realize the ecological governance of watersheds by internalizing the external cost of pollution caused by the failure of environmental governance of upstream governments [31]. Scholars in many countries have studied ecological compensation policies. Kerr [32] predicted the difficulty of managing complex watersheds based on theories from public research, arguing that while it is easier to manage at the micro-watershed level, the governance of watersheds needs to work at the macro level, and the government needs to make trade-offs between these two approaches. Aryal et al. [33] evaluated a pilot payment for ecosystem services (PES) program implemented in four different locations in Nepal. They found that the local government’s intermediary role was critical to institutionalizing the PES mechanism as a sustainable financing mechanism for securing watershed services in the country. Lu et al. [34] elaborated on the importance of protecting river systems’ ecological and hydrological linkages to maintain their healthy life cycles. They argued that China’s lack of corresponding real-time environmental regulatory data for reservoir ecological management has led to regulatory problems. Blicharska et al. [35] argued that ecological compensation is essential to stop biodiversity loss and natural values. They studied ecological compensation in Sweden and found that barriers to achieving ecological compensation were related to legislation and routines in the planning process.

### 2.2. Watershed Ecological Compensation Game

The watershed horizontal ecological compensation rules can be regarded as a sequential game. Figure 2 shows the sequential game: upstream cities are the first movers, and their governments need to make a trade-off between sacrificing productivity and receiving compensation. If the upstream city completes the ecological governance of the upstream watershed, then the upstream city is expected to receive more compensation returns. However, suppose the upstream city cannot achieve ecological governance in the short term. In that case, its government needs to make ecological compensation for the environmental pollution caused to the downstream city because their behavior has imposed external costs on the downstream city [30]. Therefore, the increase in GTFP in upstream cities is mainly due to the “compensation effect” brought about by the expectation of obtaining ecological compensation. For downstream cities, they are followers, and their governments need to formulate the best response based on the behavior of upstream cities. If the upstream cities successfully achieve ecological governance, even if the governments of the downstream cities do nothing, they still need to give the upstream cities ecological compensation. Additionally, if the downstream cities have achieved ecological governance, and the governance situation is as good as the upstream cities, then the conservative position is that the downstream cities get at least part of the ecological compensation. For downstream cities, the best response to any situation should be to choose ecological governance. Therefore, the increase in GTFP in downstream cities is mainly due to the “forced effect” of policies. The WHEC policy has successfully realized cross-government ecological governance co-operation through internalizing external costs. However, the conditions for the success of this policy require strict supervision by the central government [36]. This is because both the upstream city government and the downstream city government have the incentive to conceal their level of environmental pollution [26], to obtain ecological compensation.

### 2.3. Ecological Compensation and Green Ecological City Development

Through game analysis, this paper shows that the upstream and downstream cities of the river have intrinsic motivations to obtain ecological compensation through ecological governance, which can directly impact their green ecological development. Specifically, to achieve ecological governance, pilot cities use compensation funds to restore the watershed’s ecosystem in the pilot area in terms of source governance, for example, by returning farmland to forests and establishing nature reserves to ensure that the ecological environment at the river’s source is not damaged [37]. In terms of terminal control, enterprises that benefit from developing and utilizing natural resources need to pay a specific price for protecting and restoring the environment [38]. On the one hand, ecological fines will be imposed on enterprises that pollute rivers in the pilot city area. In contrast, ecological compensation will be granted to green production enterprises [39]. On the other hand, the governments of pilot cities will reduce the use of highly polluting energy sources and limit the production of highly polluting products [40], although this measure will also affect the employment and economic development of the city [41]. However, the pilot cities can encourage enterprises to carry out green production by obtaining ecological compensation, forming a virtuous circle. Therefore, the WHEC policy affects the improvement in the pilot city’s *GTFP* from both the source and extremity of the watershed.

In addition to the intrinsic motivation of obtaining ecological compensation to improve their green ecological development, other factors can play a moderating role on the policy’s effect in the policy-implementing cities. Firstly, technological progress is a crucial factor affecting ecological governance. Regions with advanced technology can produce higher output with lower input, reduce pollution emissions, and achieve green production [42]. Besides, technological progress is also a crucial factor for economic growth [43], and technological progress cannot be separated from R&D expenditure [44]. Therefore, a city with higher *R&D* expenditure is more able to improve its green production efficiency, and the implementation effect of the WHEC policy is also improved. In addition, to realize the ecological management of rivers, it is necessary to adjust the industrial structure of cities in the watershed. It is generally believed that secondary industry produces the most severe pollution [45,46]. After implementing the WHEC policy, the pilot city governments are expected to limit the pollution emissions of secondary industry. Nevertheless, the contribution of the secondary sector to economic development is essential [47]. Limiting pollution emissions will also restrict the output of enterprises, thereby affecting the city’s economic growth. So, in a city with an advanced industrial structure, the pollution discharge is less, and the policy effect should be greater, thereby improving the production efficiency of the city. Further, when the government attempts to manage water ecology, it must convert a lot of farmlands into forests to protect the environment. When farmers lose the land on which they live, it will hurt the farmers’ income [48]. Additionally, this releases many rural laborers and farmers who have to go to cities to seek a living, thereby speeding up urban economic development. However, this measure has exacerbated the development gap between urban and rural areas and the income gap between their respective residents [49]. Consequently, in areas with a higher income gap, farmers will be more reluctant to give up their land, thus reducing the policy’s positive effect on the area’s green ecological development. The conceptual framework can be seen in Figure 3.

Although scholars have conducted a lot of research on environmental regulation, they have mainly focused on air pollution and the overall watershed. The heterogeneity of environmental regulation in upstream and downstream watersheds has not been fully studied. In addition, the research on potential spatial correlations is insufficient. Therefore, this paper attempts to use the DID method and a spatial model to conduct spatial, heterogeneity, and mechanism analysis to supplement the research deficiencies of previous studies and provide a reference for the Chinese government to achieve the ambitious goals of green sustainable development and carbon neutrality.

## 3. Research Methodology

### 3.1. Construction of GTFP

*GTFP* was developed based on the concept of TFP. Initially, scholars only used input factors such as labor force and capital and output factors such as GDP to measure TFP [13,50]. However, TFP lacks consideration of environmental factors. Therefore, scholars have added environmental factors as unexpected outputs to measure GTFP further so that it is more helpful to evaluate the green ecological development of cities [35,51]. *GTFP* can be measured in various ways. This paper uses the Data Envelopment Analysis (DEA) method, which includes environmental pollution and other unexpected outputs, to measure the *GTFP* for multiple cities in China. The DEA method is a combination of the non-radial Slacked-Based Measure (SBM) model [52] and the Global Malmquist–Luenberger (GML) productivity index. Efficiency values of each city are more realistic when measured by the non-radial SBM model. Therefore, combining the SBM model with the GML productivity index can compensate for the defects of the original method, and the *GTFP* measured by the SBM–GML method with unexpected output is more comprehensive and reasonable.
(1)$$\vec{D_{0}^{t}}=\left(x^{t},y^{t},b^{t}\right)=\hat{p}=min\frac{1-\frac{1}{N}\sum_{n=1}^{N}\frac{s_{n}^{-}}{x_{ni}}}{1+\frac{1}{D+J}\left(\sum_{d=1}^{D}\frac{s_{d}^{+}}{y_{di}}+\sum_{j=1}^{J}\frac{s_{j}^{b-}}{b_{ji}}\right)}$$
(2)$$\mathrm{s.t.}\left\{\begin{array}{l} P_{t}\left(x\right)=\left\{\left(y_{t},b_{t}\right):x_{t}\;can\;produce_{}\left(y_{t},b_{t}\right),X\in R_{+}^{N},Y\in R_{+}^{D},B\in R_{+}^{J}\right\}\\ \sum_{i=1}^{I}z_{i}^{t}y_{id}^{t}-s_{d}^{+}=y_{id}^{t},d=1,\ldots ,D\\ \sum_{i=1}^{I}z_{i}^{t}b_{ij}^{t}+s_{j}^{b-}=b_{ij}^{t},j=1,\ldots ,J\\ \sum_{i=1}^{I}z_{i}^{t}x_{in}^{t}+s_{n}^{+}=x_{in}^{t},n=1,\ldots ,N\\ \sum_{i=1}^{I}z_{i}^{t}=1,z_{i}^{t}\geq 0,s_{d}^{+}\geq 0,s_{j}^{b-}\geq 0,s_{n}^{+}\geq 0,i=1,\ldots ,I \end{array}\right.$$

Measuring *GTFP* requires the continuous increase in expected output and the constant decrease in unexpected output. The GML productivity index (our *GTFP* measure) helps to measure the dynamic changes in production efficiency levels more objectively. Therefore, the $GTFP_{t}^{t+1}$ index change of the Chinese cities changing from period *t* to period *t* + 1 is expressed as:(3)$$GTFP_{t}^{t+1}={\left\{\frac{1+\vec{D_{0}^{t}}\left(x^{t},y^{t},b^{t}\right)}{1+\vec{D_{0}^{t}}\left(x^{t+1},y^{t+1},b^{t+1}\right)}\times \frac{1+\vec{D_{0}^{t+1}}\left(x^{t},y^{t},b^{t}\right)}{1+\vec{D_{0}^{t+1}}\left(x^{t+1},y^{t+1},b^{t+1}\right)}\right\}}^{\frac{1}{2}}$$

The *GTFP* can be decomposed into Efficiency Change (*EC*) and Technological Progress Change (*TC*). *EC* reflects the efficiency distance of cities, while *TC* measures the degree of technological progress. Equations (4)–(6) show the decomposition of *GTFP* as follows:(4)$$GTFP_{t}^{t+1}=EC_{t}^{t+1}\times TC_{t}^{t+1}$$
(5)$$EC_{t}^{t+1}=\frac{1+\vec{D_{0}^{t}}\left(x^{t},y^{t},b^{t}\right)}{1+\vec{D_{0}^{t}}\left(x^{t+1},y^{t+1},b^{t+1}\right)}$$
(6)$$TC_{t}^{t+1}={\left\{\frac{1+\vec{D_{0}^{t+1}}\left(x^{t},y^{t},b^{t}\right)}{1+\vec{D_{0}^{t}}\left(x^{t+1},y^{t+1},b^{t+1}\right)}\times \frac{1+\vec{D_{0}^{t+1}}\left(x^{t},y^{t},b^{t}\right)}{1+\vec{D_{0}^{t}}\left(x^{t+1},y^{t+1},b^{t+1}\right)}\right\}}^{\frac{1}{2}}$$

In Equations (1)–(6), *x_ni_* is the amount of input, *y_di_* is the expected output, *b_ji_* is the unexpected output, $s_{n}^{-}$, $s_{d}^{+}$ and $s_{j}^{b-}$ the amount of slack input in various industry elements, expected to be produced and unexpected to be produced. *I* represents the number of decision-making units, *n* represents the input quantity of each decision-making unit, *d* represents the expected output quantity of each decision-making unit, *j* represents the undesired output quantity of each decision-making unit, and *i* is used to distinguishing each city. *GTFP* evaluates the dynamic changes in the green ecological development level of various cities. Due to the difference in multiple cities of China, the variable return to scale (VRS) method was used to calculate the target value of *GTFP_t_*^*t*+1^ for each city in the current year, *EC_t_*^*t*+1^ and *TC_t_*^*t*+1^, which are derived from decomposition. If the *GTFP_t_*^*t*+1^ is greater than 1, it means that the city’s *GTFP* has improved from the previous year. Likewise, a *GTFP* of less than 1 indicates declining green ecological development. As for *EC* and *TC*, an *EC* more significant than 1 means a reasonable allocation optimization of various inputs and outputs of the city and a *TC* greater than 1 indicates that the city’s technology has improved and vice versa, respectively.

This paper uses the data from 27 cities in China from 2006 to 2019 to measure *GTFP*. Given that 2006 was used as the base year for *GTFP* calculation (which means the year 2006 will be ignored in the result), the estimated results cover the periods from 2007 to 2019. The actual capital stock, average number of employees, municipal district area, and energy consumption of each city are inputs. The real capital stock is estimated using the perpetual inventory method as follows:(7)$$K_{t}=\left(1-\delta_{t}\right)\times K_{t-1}+I_{t}÷FAI_{t}$$
where *K_t_* is the fixed asset of each year, *K_t−1_* is the fixed asset of the previous year, and *FAI_t_* is the price index of investment in fixed assets based on 2006. *δ_t_* is the annual depreciation rate. Energy consumption considers all types of energy and is converted into 10,000 tons of standard coal with an energy conversion factor. As for outputs, the real *GDP* (year 2006 as the base year) was the expected output. The SO_2_ emissions (tons), industrial wastewater discharge (10,000 tons), and smoke dust emissions (tons) were unexpected outputs.

### 3.2. Assessing the Impact of WHEC Policy—DID Method

The implementation of the WHEC policy can be regarded as a quasi-experiment, and this paper used the DID (difference-in-difference) method proposed by Heckman et al. [53] to assess the impact of the WHEC policy on *GTFP*. The differences between the groups can accurately identify the policy effects of the WHEC policy on *GTFP*. The DID regression equation can be established as follows:(8)$$GTFP_{it}=\mu_{0}+\mu_{1}{did}_{it}+\beta X_{it}+\omega_{i}+\pi_{t}+\delta_{it}$$

In Equation (8), *i* represents the city, *t* represents the year, *GTFP_it_* represents the green total factor productivity of the *i*-th city in the *t*-th year, and *δ* is random error. *did_it_* is a dummy variable, which is 1 after the policy implementation year (2011) in Huangshan City and Hangzhou City. Otherwise, it is 0. *μ*_3_ represents the net effect of implementing WHEC on the treated group’s mean *GTFP*. Since this study is based on panel data, the city-fixed effects (*ω_i_*) and year-fixed effects (*π_t_*) were considered while adding city-level control variables (*X_it_*).

*X_it_* represents a series of covariates that affect *GTFP* including: (1) *R&D* intensity (*R&D*), which is measured by the ratio of *GDP* to *R&D* internal expenditure. The larger the *R&D*, the smaller the *R&D* intensity, and the *R&D* incentives can improve technological progress [54], thereby increasing the city’s *GTFP*. (2) Industrial structure (*IS*), which is measured by the ratio of the added value of the primary industry to the secondary industry. The larger the *IS*, the lower the degree of industrial upgrading, and the more the industrial structure upgrades can reduce the proportion of polluting industries [55]. (3) Fiscal general budget expenditure (*Be*), which is measured by the general budget expenditure. Government fiscal expenditures can encourage enterprises to carry out more green production [56], thereby increasing the city’s *GTFP*. (4) Education level (*Edu*), which is measured by the number of university students. The higher the overall education level of the city, the more emphasis there is on the concept of ecological environmental protection [57] and the easier it is to obtain human capital; (5) Transportation convenience (*Tran*), which is measured by the total number of passengers transported. The more convenient the city’s transportation, the more convenient it is to obtain input factors such as labor force, and the higher the value of output factors such as *GDP* [58], thereby increasing the city’s *GTFP*. (6) Internet development (*Inter*) which is measured by the total telecom business. A higher level of internet development in a city can improve the city’s digitalization process, thereby increasing the city’s productivity [59]. (7) Electricity consumption (*Ec*), which is measured by the annual electricity consumption. Electricity consumption is an integral part of energy consumption, and energy consumption is an essential factor affecting the city’s *GTFP* [60]. All measures are expressed in logarithms except the variables *R&D* and *IS*.

This paper further examines the spatial effects of the WHEC policy and the moderating effects of *R&D* intensity, industrial structure, and income gap as mechanisms. It is important to note that spatial dependence can affect the *GTFP* and that different *R&D* intensity levels, industrial structure and income gap between cities may influence the impact of the WHEC policy.

### 3.3. Data Sources

Data were obtained from various sources, namely, the “China City Statistical Yearbook”, “China Energy Statistical Yearbook”, “China Science and Technology Statistical Yearbook”, “Anhui Statistical Yearbook”, and “Zhejiang Statistical Yearbook” (Data sources for the yearbook are referenced on the following website: https://data.cnki.net/yearBook?type=type&code=A (accessed on 15 August 2022)). The data covers 27 cities from 2007 to 2019, with 2 cities implementing the Watershed Horizontal Ecological Compensation policy and 25 cities that did not.

## 4. Results and Discussion

### 4.1. Descriptive Statistics

The implementation of the policy took place in 2011. Table 1 reports the descriptive statistics. The mean value of *GTFP* in the treated group was 1.01, which was slightly higher than the mean value of *GTFP* in the control group, indicating that the overall green development degree of the cities in the treated group was higher than that in the control group. This preliminarily proves the effectiveness of the WHEC policy in improving *GTFP*. Overall, these cities have a high degree of greening because the average value of their *GTFP* is at least 1. In addition, examining the decomposition of the GTFP indicates that the mean value of *TC* in the treated group (1.01) seems slightly higher than in the control group (1), while the *EC* of both the treated and control groups is 1. Overall, the assessment of the decomposition values indicates that TC dramatically contributes to the improvements in GTFP more than *EC*.

### 4.2. The State of GTFP and its Distribution

Figure 4 reports the *GTFP* scores for each city in 2010 (before policy implementation) and 2011 after policy implementation. After the implementation in 2011, the GTFP of the two cities in the treated group showed different changes. The *GTFP* of Hangzhou (the downstream pilot city) showed different degrees of improvement after implementing the policy, which preliminarily verified the positive impact of the WHEC policy. In contrast, the *GTFP* of Huangshan City (the upstream pilot city) began to decline gradually after the implementation of the policy, and showed a fluctuating rise in the later period (Figure 5), indicating that the WHEC policy may produce differences in the improvement in the GTFP of upstream and downstream cities. However, the average treatment effects of the treated group require proper evaluation, and the DID method would be more appropriate than a visual inspection of the *GTFP*.

### 4.3. Impact of Environmental Regulations—DID Results

Table 2 reports the regression results of the basic DID as specified in Equation (8). For comparative analysis, we show three models. In column (1), without the other covariates and fixed effects, the coefficient of *did* is significantly positive at the 5% level, indicating that the WHEC policy positively impacts *GTFP*. In column (2), which does not control the city-fixed effect, the coefficient of *did* is still positive at the 5% level. By examining the complete model controlling the city fixed effect and year fixed effect, as column (3) indicates, adding other covariates results in a significant impact of the WHEC policy at a 5% level, we can observe that the WHEC policy can significantly improve the green ecological development of the pilot city [26,61]. However, the coefficient of *did* increased from 0.0264 to 0.0291, indicating that other control variables also have an impact on the cites’ GTFP.

### 4.4. Mechanism Assessment and Regional Heterogeneity

This study constructed a moderator effect model to test the moderating effect of the WHEC policy on the cities’ *GTFP*. The basis is to assess whether the WHEC policy could affect the cities’ *GTFP* through a mechanism, namely innovation, industrial structure, and income gap. The equations are as follows:(9)$$GTFP_{it}=\mu_{0}+\mu_{1}{did}_{it}*M_{it}+\beta X_{it}+\omega_{i}+\pi_{t}+\delta_{it}$$

*M_it_* represents the moderator variable. This study selected *R&D* incentives (*R&D*) and industrial structure (*IS*), and Theil index (*TI*) as moderator variables based on the literature. Referring to the calculation of Cheng et al. [62], the equation of the Theil index is as follows:(10)$$TI_{it}=\sum_{j=1}^{2}\left(\frac{p_{jt}}{p_{t}}\right)\ln\left(\frac{p_{jt}}{p_{t}}÷\frac{z_{jt}}{z_{t}}\right)$$

In Equation (10), *j* = 1 and 2 represent urban and rural areas, respectively, *z_jt_* represents the population, *z_t_* represents the total population in year *t*, *p_jt_* represents total income (measured by the product of population and per capita income), and *p_t_* represents total income in year *t*. *TI* represents the Theil index. The larger the *TI*, the more significant the income gap. The remaining variables are the same as in the previous Equation (8). To distinguish the heterogeneity of policies in the upstream and downstream pilot cities, this study divided the sample into two sub-samples of Zhejiang Province and Anhui Province and re-tested.

Table 3 and Table 4 report the regression results for the mechanism analysis and subsamples, respectively. The overall sample results show that the coefficients of *did*R&D* and *did*IS* are both significantly negative, and as the sample sizes of *R&D* and *IS* are more extensive, the coefficients of the interaction term are more minor, indicating that the WHEC policy is more effective in cities with higher *R&D* incentives and more upgraded industrial structure. Interestingly, the income gap did not reflect the expected moderating effect. The possible reason for this is that China’s targeted poverty alleviation policy has reduced the income gap between urban and rural areas [63,64], resulting in a negative though insignificant coefficient for the *did*TI*. In addition, the impact of the WHEC policy on *GTFP* varies by region. The positive effect of this policy on the *GTFP* of downstream pilot cities (0.045) is higher than that of upstream pilot cities (0.0183). It proves that the improvement in the *GTFP* of upstream cities by the WHEC policy is more likely to be realized through compensatory effects, while for downstream cities it may be more likely to be realized through “force effects”, and the “forcing effect” of the policy is higher than the “compensating effect”.

### 4.5. Spatial Analysis

Spatial effects due to spatial dependence can play a crucial role. Therefore, this paper further examines the spatial impact of the WHEC policy on *GTFP*. This paper uses *Moran’s I* index calculated from the inverse geographic distance matrix to test whether *GTFP* has a spatial effect. The specific equation is as follows:(11)$$W1_{ij}=\left(\begin{array}{ccc} 0 & w\left(d_{ji}\right) & w\left(d_{ik}\right)\\ w\left(d_{ij}\right) & 0 & w\left(d_{jk}\right)\\ w\left(d_{ki}\right) & w\left(d_{kj}\right) & 0 \end{array}\right)\;\;\;w\left(d_{ij}\right)=\frac{1}{d_{ij}}$$

The economic distance matrix can also be further constructed for robustness,
(12)$$W2_{\mathrm{ij}}=W1*\mathrm{diag}\left(\frac{\bar{GDP_{1}}}{\bar{GDP}},\frac{\bar{GDP_{2}}}{\bar{GDP}},\ldots .\frac{\bar{GDP_{n}}}{\bar{GDP}}\right)$$
(13)$$\mathrm{Global}\;Moran’s\;I=\frac{\sum_{i=1}^{n}\sum_{j=1}^{n}W_{ij}\left(Y_{i}-\bar{Y}\right)\left(Y_{j}-\bar{Y}\right)}{S^{2}\sum_{i=1}^{n}\sum_{i=1}^{n}W_{ij}}$$

In Equations (11)–(13), *W_ij_* is the spatial weight matrix, *S*^2^ is the sample variance, *Y* is the sample mean. Based on Equation (8), this paper constructs the spatial autoregressive model (SAR) to study the possible spatial effects of the WHEC policy. The SAR model is as follows:(14)$$GTFP_{\mathrm{it}}=\rho_{0}+\delta *W{*\mathrm{GTFP}}_{\mathrm{it}}+\mu_{1}{\mathrm{did}}_{\mathrm{it}}+\beta X_{\mathrm{it}}+\omega_{i}+\pi_{t}+\nu_{\mathrm{it}}$$

In Equation (14), *W* is the spatial weight matrix. The remaining variables are the same as in the previous Equation (8).

Table 5 reports the results of *GTFP* spatial autocorrelation. From 2007 to 2019, *Moran’s I* Index was only significant for four years, which shows that *GTFP* has a spatial correlation. Table 6 reports the regression results of the spatial impact of the WHEC policy on *GTFP*. No matter which spatial weighting matrix is used, it shows that the WHEC policy has a significant promoting effect on the improvement in *GTFP*. In addition, the spatial coefficient of *GTFP* is negative while not significant, indicating that the progress of *GTFP* in pilot cities has a potential inhibitory effect on the improvement in *GTFP* in surrounding cities. The possible reason is that the polluting industries in pilot cities are transferring to cities that have not implemented the policy [65,66]. Therefore, non-pilot cities may become “pollution heaven”.

### 4.6. Robustness Test

We further conducted the robustness test to confirm the reliability of the estimated results by using the PSM-DID method, changing the policy implementation time and pilot cities, and imposing alternative indicators for *GTFP*. The following section reports the approach and results.

#### 4.6.1. Parallel Trend Test

The fundamental assumption of the DID method is the parallel trend, where the treated and control groups must have a parallel trend before the policy is implemented and cannot change significantly over time. The regression equation is as follows,
(15)$$GTFP_{\mathrm{it}}=\alpha_{0}+\sum_{j=-m}^{n}\alpha_{j}{\mathrm{did}}_{i,t-m}+\beta_{r}X_{\mathrm{it}}+\omega_{i}+\pi_{t}+\lambda_{it}$$
where *did* is a dummy variable. If city *i* has implemented the policy during the *t-j* period, then the value of this variable is 1. Otherwise, it is 0. *m* and *n,* respectively represent the number of periods before and after the policy implementation. Regarding scholars’ research results [67,68], the first year (2010) before the policy implementation was removed to prevent multicollinearity.

Figure 6 reports the results of the parallel trend test, showing that the treated and control groups had the same trend before 2011, which supports the parallel trend hypothesis and verifies the rationality of the DID method. Overall, the WHEC policy has a positive effect on improving the city’s *GTFP*.

#### 4.6.2. PSM–DID

The PSM–DID (difference-in-differences based on propensity matching scores) method combines the advantages of the PSM and the DID methods, which can eliminate the errors caused by endogenous selection and facilitate the control of unobservable variables that do not change with time. While DID can provide reliable estimates, it is essential to note that simply comparing the differences in the outcome variables in the treated and control groups may lead to biased estimators, given that observational variables in the two groups may differ. The PSM method compares the treated group by finding samples as similar as possible to the treated group to form a control group (matching). We performed a regression on the matched data as specified in Equation (8). When results in Table 7 were compared with the results in Table 2, the sign and significance of the coefficient for *did* have not changed significantly, indicating that the results are robust. Specifically, after controlling for the time and city fixed effects, the coefficient of *did* rose from 0.0291 to 0.0310, indicating that after finding a more similar control group through matching, the positive impact of the WHEC policy on cites’ *GTFP* further increased.

#### 4.6.3. Exclude Other Policies Test

GTFP can be affected by many other macro-policies. This paper tested robustness by considering two additional types of policy. One is the “Five-Year Plan” implemented by the Chinese government. Since the selected sample time includes “The Twelfth Five-Year Plan (2011–2015)” and “The Thirteenth Five-Year Plan (2016–2020)”, this paper identifies them as two macro-policies. The second is the environmental protection tax policy. China began to collect environmental protection tax across the country on 1 January 2018. Since these two types of policy will have an impact on the city’s *GTFP*, to exclude the impact of these two macro-policies, this paper constructs the following equation according to Equation (8):(16)$$GTFP_{\mathrm{it}}=\mu_{0}+\mu_{1}{\mathrm{did}}_{\mathrm{it}}+\mu_{2}\mathrm{did}1_{\mathrm{it}}+\mu_{3}\mathrm{did}2_{\mathrm{it}}+\mu_{4}\mathrm{did}3_{\mathrm{it}}+\beta X_{\mathrm{it}}+\omega_{i}+\pi_{t}+\delta_{\mathrm{it}}$$

In Equation (16), *did*1, *did*2 and *did*3 are dummy variables. If city *i* implements the policy in year *t*, then *did*1–3 is 1 after the *t*-th year, and 0 otherwise. Table 8 reports the estimated results after controlling for these policies. Regardless of whether a single or two types of policy are controlled, the coefficient of *Time*policy* is still significantly positive at the 5% level, indicating that the increase in *GTFP* is caused by the WHEC policy studied in this paper.

#### 4.6.4. Placebo Test

This paper uses two approaches to test for placebo effects. First, to judge whether the benchmark result is robust, we changed the time of policy implementation and checked if there was a significant change in the coefficient of *did* [69]. For this, the first year and the fourth year after the policy implementation year (2011) were taken as false policy implementation years. If the coefficient of the *did* is still significantly positive, it indicates that other existing factors have caused a significant difference in the *GTFP*. Otherwise, it proves the robustness of the DID regression results. Table 7 shows the results of the time placebo test, which is not significant. The second method involved randomization of the treated group and control group. This paper randomly selected two cities as a “pseudo-treated group” from 27 cities and then generated “pseudo-policy dummy variables”. Then, we re-tested Equation (8) by simulating it approximately 1000 times, which can draw 1000 estimated coefficients and their corresponding *p*-values. Figure 7 shows the results of the placebo test. The vertical dotted line is the real estimation value of the DID model 0.0291 (Table 2), and the horizontal dotted line is a significant level at 0.1. Figure 7 shows that most estimated coefficients are concentrated near zero, and most *p*-values are greater than 0.1. This means that the improvement in the *GTFP* in the treated cities is not caused by other random factors but caused by the WHEC policy.

#### 4.6.5. Surrogate Index Test

To avoid the randomness of index selection affecting the regression results, this study used three different methods to recalculate *GTFP*, namely SBM–DDF (*GTFP*1), GML–DDF (*GTFP*2), and EBM–GML (*GTFP*3). Then, it used them to re-test the DID model. The results show that no matter which method was used to calculate *GTFP* as the explained variable, the estimated coefficient of *did* was positive and statistically significant (see Table 9), further confirming the robustness of the positive effect of the WHEC policy on the green ecological development of the city.

## 5. Conclusions

This paper aims to evaluate *GTFP* across time and space. Using the data covering 27 cities in Anhui and Zhejiang provinces from 2007 to 2019, it analyses the direct and spatial impact of the Watershed Horizontal Ecological Compensation policy on *GTFP*. The paper further examines the moderation effect of *R&D* incentives, industrial structure and income gap. Firstly, the results show that the *GTFP* of Zhejiang and Anhui cities is generally higher, indicating that these cities have a higher degree of greening. Moreover, after implementing the WHEC policy in 2011, the *GTFP* of the treated groups all improved to varying degrees, indicating that applying the theory of common property rights to cross-watershed ecological governance can solve the problem of pollution externalities in upstream and downstream watersheds. However, this promotion effect showed heterogeneity in the upstream and downstream pilot cities. The increase in *GTFP* in the downstream pilot cities is due to the “force effect”, while the increase in *GTFP* in the upstream pilot cities is due to a “compensatory effect”. Additionally, the policy had a better effect on improving *GTFP* in the cities with higher *R&D* incentives and industrial structure upgrades. In the case of cities with a greater income gap, there was no significant moderating effect on the policy effect due to the impact of China’s targeted poverty alleviation policy. Interestingly, the increase in *GTFP* in the pilot cities that implement the WHEC policy will potentially inhibit the increase in *GTFP* in the cities that do not, which will easily lead to the problem of a “pollution heaven”. Our results are robust despite re-running estimates using different periods and proxies. Therefore, policymakers need to strengthen the promotion effect of this policy on *GTFP* through industrial structure and technological progress and cooperate with targeted poverty alleviation policies to further weaken its negative impact. Improving the city’s *GTFP* for long-term sustainability requires the government to further recognize the limitations and heterogeneity of policy effects, ensure inter-governmental coordination and co-operation, and incorporate more potential entrants into the policy to achieve green and sustainable urban development.

Based on the above conclusions, this paper proposes the following policy recommendations to promote further China’s watershed ecological governance and urban green ecological development. The first is to promote the implementation of cross-watershed ecological governance policies on a larger scale. Meanwhile, it is important to pay attention to the central government’s supervision of the ecological environment of local governments to avoid the failure of cross-watershed ecological governance policies caused by local governments’ concealment of information. Second, policymakers should adapt to local conditions when implementing cross-watershed ecological governance policies and carry out targeted ecological governance based on the conditions of different watersheds and regions. Third, the government should actively improve *R&D* incentives and industrial structure upgrades to enhance the positive effect of cross-watershed ecological governance policies on urban green ecological development and, at the same time, continue to implement poverty alleviation policies to reduce the external effects of watershed ecological governance policies. Fourth, the government should pay attention to the potential pollution spillover, encourage more regions to join the cross-basin ecological governance policy, and prevent the surrounding areas from becoming a “pollution heaven” because of the lack of ecological governance.

The main limitations of this paper lie in the following three aspects. First, based on the panel data of 27 cities in the Anhui and Zhejiang provinces, this study found that the WHEC policy positively affects the green ecological development of these 27 cities. However, policy implementation needs to be tailored to local conditions. Therefore, future research can focus on comparing with the policies of watershed ecological governance in different countries to improve the relevant theories of watershed ecological governance. Second, to avoid the impact of COVID-19, the panel data in this article are only updated until 2019. Regrettably, during the COVID-19 period, the government may have lacked supervision and trade-offs between economic development and environmental protection, affecting policy effects. Therefore, longer-term panel data can be considered in future studies to capture policy effects during COVID-19 more accurately. Third, there may still be unknown mechanisms for regulating watershed ecological governance policies for urban green ecological development. Therefore, in future research, it can be considered to further expand the conceptual framework of this study by combining theories to identify a complete mechanism of action.

## Figures and Tables

**Figure 1 ijerph-20-02679-f001:**
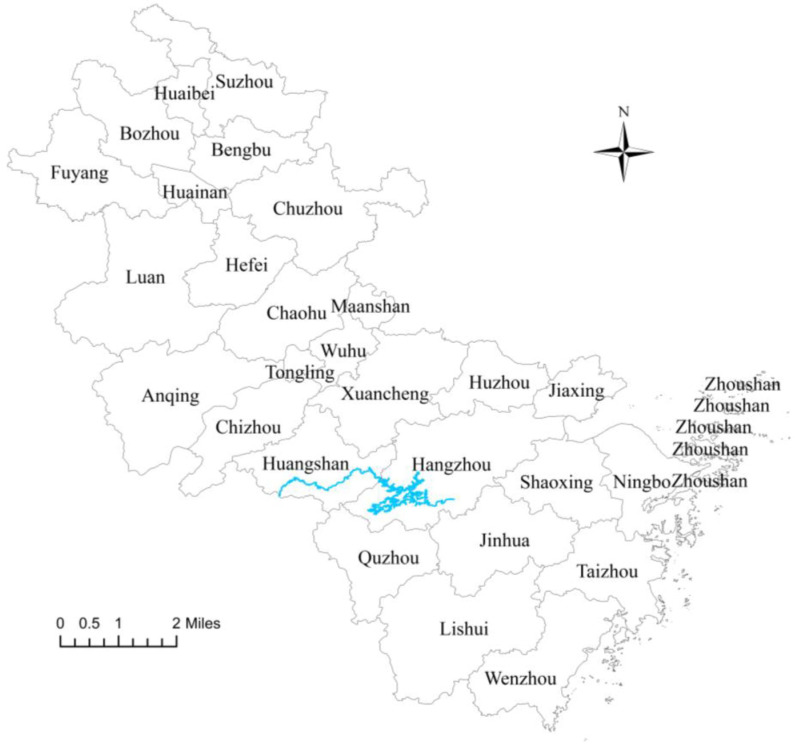
Anhui, Zhejiang Provinces and Xin’an watershed. The blue/highlighted part represents the Xin’an watershed. Chaohu City was merged into Hefei City.

**Figure 2 ijerph-20-02679-f002:**
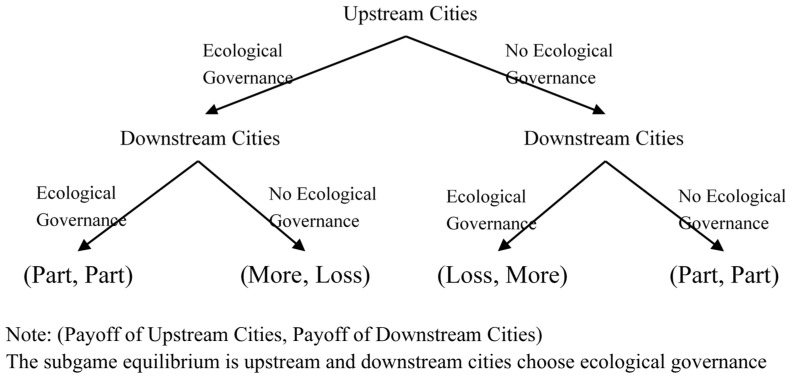
The Game of Upstream and Downstream Cities.

**Figure 3 ijerph-20-02679-f003:**
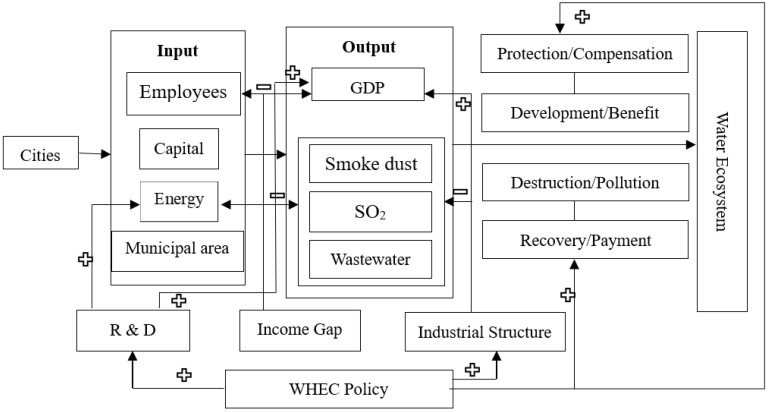
Conceptual Framework of WHEC Effect on GTFP.

**Figure 4 ijerph-20-02679-f004:**
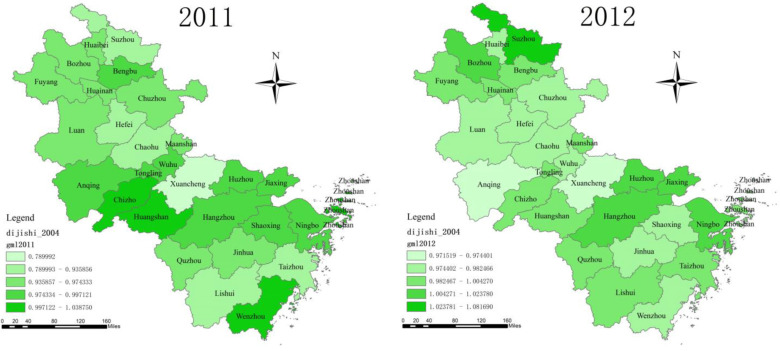
GTFP Across Zhejiang and Anhui Cities, 2010 and 2011. Note: Chaohu City was merged into Hefei City.

**Figure 5 ijerph-20-02679-f005:**
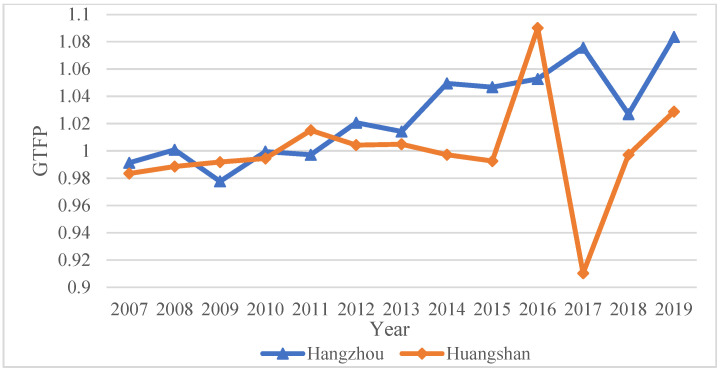
GTFP Trend in Huangshan and Zhejiang city from 2007 to 2019.

**Figure 6 ijerph-20-02679-f006:**
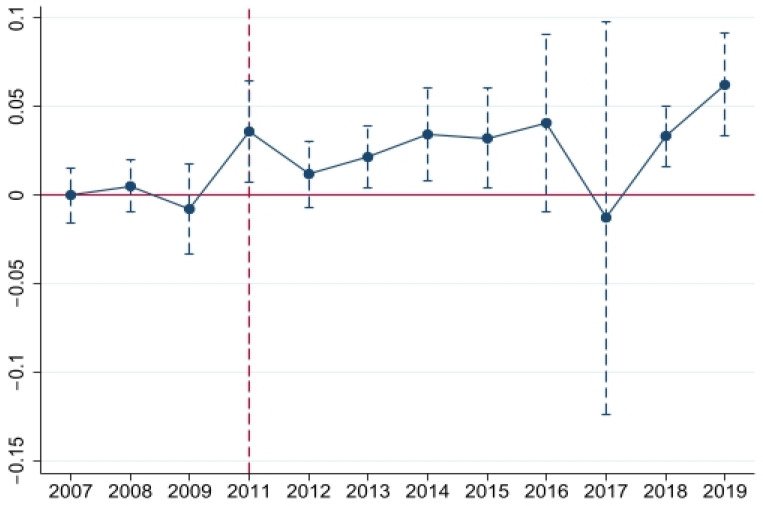
Parallel Trend Test Results.

**Figure 7 ijerph-20-02679-f007:**
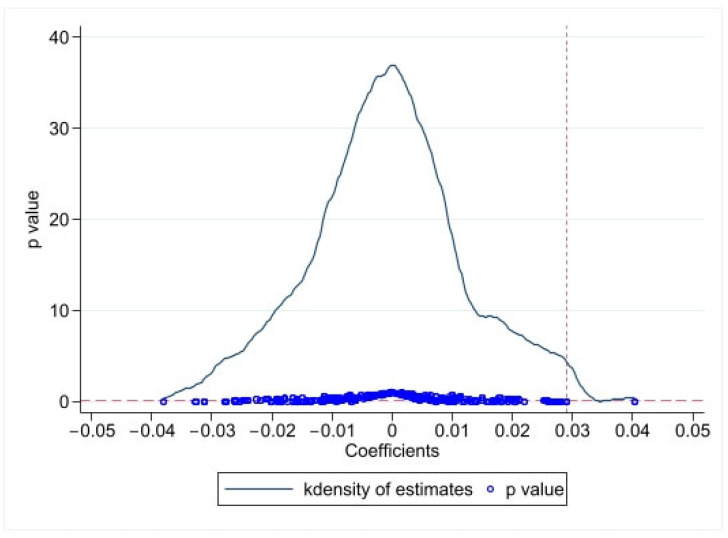
Placebo Test Result.

**Table 1 ijerph-20-02679-t001:** Descriptive Statistics.

Variable	Treated Group					Control Group				
	N	Mean	SD	Min	Max	N	Mean	SD	Min	Max
*GTFP*	26	1.010	0.040	0.910	1.090	325	1	0.030	0.790	1.110
*EC*	26	1	0.020	0.950	1.070	325	1	0.040	0.720	1.310
*TC*	26	1.010	0.040	0.910	1.090	325	1	0.030	0.760	1.300
*R&D*	26	53.20	23.44	28.98	116.6	325	86.81	71.35	10.34	338.4
*IS*	26	0.170	0.110	0.070	0.390	325	0.240	0.200	0.020	1.080
*Be*	26	14.94	1.200	12.73	16.79	325	14.57	0.780	12.47	16.69
*Edu*	26	11.38	1.610	9.490	13.07	325	10.60	0.900	8.010	13.18
*Tran*	26	9.300	1.040	7.660	10.50	325	9.110	0.710	7.280	10.60
*Inter*	26	13.02	1.310	11.41	14.52	325	12.44	0.890	9.420	14.60
*Ec*	26	13.99	1.840	11.71	15.92	325	13.98	0.840	12.34	15.90

**Table 2 ijerph-20-02679-t002:** DID Estimation Results.

Variables	(1)	(2)	(3)
*did*	0.0264 **	0.0264 **	0.0291 **
	(0.0124)	(0.0126)	(0.0106)
*Control variable*	No	No	Yes
*Constant*	0.9917 ***	0.9899 ***	0.6362 **
	(0.0018)	(0.0053)	(0.2639)
*City-fixed effect*	No	No	Yes
*Year-fixed effect*	No	Yes	Yes
*N*	351	351	351
*R* ^2^	0.0213	0.2665	0.2945

Note: The value in brackets is the standard error of clustering at the city level. *** indicates *p* < 0.01 and ** indicates *p* < 0.05.

**Table 3 ijerph-20-02679-t003:** Results of Mechanism Analysis.

Variables	(1) *GTFP* *M = R&D*	(2) *GTFP* *M = IS*	(3) *GTFP* *M = TI*
*did*	0.0590 ***	0.0576 ***	0.0483 **
	(0.0172)	(0.0058)	(0.0260)
*did*M*	−0.0005 **	−0.1832 ***	−0.4127
	(0.0002)	(0.0378)	(0.4508)
*did*25%M*	0.0576 ***	0.0546 ***	0.0441
	(0.0172)	(0.0063)	(0.0311)
*did*50%M*	0.0569 ***	0.0525 ***	0.0415
	(0.0172)	(0.0072)	(0.0357)
*did*75%M*	0.0556 ***	0.0473 ***	0.0395
	(0.0173)	(0.0104)	(0.0396)
*Control variable*	Yes	Yes	Yes
*City-fixed effect*	Yes	Yes	Yes
*Year-fixed effect*	Yes	Yes	Yes
*Constant*	0.5703 **	0.5484 **	0.3526
	(0.2579)	(0.2520)	(0.4163)
*N*	351	351	270
*R* ^2^	0.2979	0.2982	0.3437

Note: In column (3), the Theil index is added as a control variable, and the Theil index was only calculated for 2010 and later due to missing data, so the *N* is 270. The value in brackets is the standard error of clustering at the city level. *** indicates *p* < 0.01 and ** indicates *p* < 0.05.

**Table 4 ijerph-20-02679-t004:** Results of Heterogeneity Analysis.

Variable	*GTFP*	
	Anhui Province	Zhejiang Province
*did*	0.0183 ***	0.0451 ***
	(0.0054)	(0.0071)
*Control variable*	Yes	Yes
*City-fixed effect*	Yes	Yes
*Year-fixed effect*	Yes	Yes
*Constant*	0.4975	0.7007
	(0.3974)	(0.6385)
*N*	207	144
*R* ^2^	0.2823	0.4595

Note: The value in brackets is the standard error of clustering at the city level. *** indicates *p* < 0.01.

**Table 5 ijerph-20-02679-t005:** Spatial Autocorrelation Test Results.

Year	*Global Moran’s I*	*z*
2007	0.029 **	1.989
2008	0.014 *	1.709
2009	0.030 **	1.957
2010	−0.037	0.039
2011	−0.055	−0.472
2012	−0.049	−0.296
2013	−0.052	−0.338
2014	−0.018	0.515
2015	0.002	0.997
2016	−0.027	0.275
2017	−0.006	0.817
2018	0.205 ***	6.224
2019	0.022	1.548

Note: *** indicates *p* < 0.01, ** indicates *p* < 0.05, and * indicates *p* < 0.1.

**Table 6 ijerph-20-02679-t006:** Spatial Regression Results.

Variables	(1) *W = W*1	(2) *W = W*2
*did*	0.0291 ***	0.0290 ***
	(0.0103)	(0.0102)
*δ*W*GTFP*	−0.1275	−0.0736
	(0.1459)	(0.1330)
*Control variable*	Yes	Yes
*City-fixed effect*	Yes	Yes
*Year-fixed effect*	Yes	Yes
*N*	351	351
*R* ^2^	0.0892	0.0891

Note: The value in brackets is the standard error of clustering at the provincial level. *** indicates *p* < 0.01.

**Table 7 ijerph-20-02679-t007:** PSM–DID Estimation Results.

Variables	(1)	(2)	(3)
*did*	0.0253 *	0.0248 *	0.0310 ***
	(0.0133)	(0.0142)	(0.0067)
*Control variable*	No	No	Yes
*Constant*	0.9906 ***	1.003 ***	0.9373 **
	(0.0043)	(0.0079)	(0.4229)
*City-fixed effect*	No	No	Yes
*Year-fixed effect*	No	Yes	Yes
*N*	141	141	141
*R* ^2^	0.0508	0.3908	0.4325

Note: The value in brackets is the standard error of clustering at the city level. *** indicates *p* < 0.01, ** indicates *p* < 0.05, and * indicates *p* < 0.1.

**Table 8 ijerph-20-02679-t008:** DID Estimation Results—Exclude Other Policies.

Variables	(1)	(2)	(3)
*did*	0.0290 **	0.0290 **	0.0290 **
	(0.0105)	(0.0105)	(0.0105)
*did* _1_	Yes		Yes
*did* _2_		Yes	Yes
*did* _3_		Yes	Yes
*Control variable*	Yes	Yes	Yes
*Constant*	0.6362 **	0.6362 **	0.6362 **
	(0.2639)	(0.2639)	(0.2639)
*City fixed effect*	Yes	Yes	Yes
*Year fixed effect*	Yes	Yes	Yes
*N*	351	351	351
*R* ^2^	0.2945	0.2945	0.2945

Note: The value in brackets is the standard error of clustering at the city level. ** indicates *p* < 0.05. *did*1 represents the environmental tax policy, *did*2 is “The Twelfth Five-Year Plan”, and *did*3 is “The Thirteenth Five-Year Plan”.

**Table 9 ijerph-20-02679-t009:** Robustness Test Estimates.

Variables	(1) First Year	(2) Fourth Years	(3) *GTFP*1	(4) *GTFP*2	(5) *GTFP*3
*did*	0.0207	0.0184	0.0301 **	0.0249 ***	0.0344 **
	(0.0138)	(0.0116)	(0.0116)	(0.0058)	(0.0139)
*Control variable*	Yes	Yes	Yes	Yes	Yes
*Constant*	0.6347 **	0.6311 **	0.4368	0.4371 **	1.0024 ***
	(0.2598)	(0.2604)	(0.3396)	(0.4559)	(0.1993)
*City fixed effect*	Yes	Yes	Yes	Yes	Yes
*Year fixed effect*	Yes	Yes	Yes	Yes	Yes
*N*	351	351	351	351	351
*R* ^2^	0.2891	0.2873	0.0880	0.1264	0.2245

Note: The value in brackets is the standard error of clustering at the city level. *** indicates *p* < 0.01 and ** indicates *p* < 0.05.

## Data Availability

The data presented in this study are available on request from the first author. The data are not publicly available due to privacy.

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
