# Peer review of "Watershed Horizontal Ecological Compensation Policy and Green Ecological City Development: Spatial and Mechanism Assessment"

_ijerph, 2023, doi:10.3390/ijerph20032679_

Round 1

Reviewer 1 Report

1. In the introduction, you can add secondary data related to the gap phenomenon

2.Add suggestions for further research at the conclusion

Author Response

Comments and Suggestions for Authors

  1. In the introduction, you can add secondary data related to the gap phenomenon

Re: Thank you for your comments. However, there are some empirical studies on the secondary data of the ecological compensation policy in the Xin'an River, so maybe this inappropriate statement is a gap phenomenon, which has been expressed separately, see section 1

“After the policy was put forward, although many scholars have done research on watershed ecological governance (Sheng and Han, 2022; Wan et al., 2022), they have not discussed the heterogeneity of the policy effects of ecological governance in upstream and downstream cities, and the research on the potential spatial correlation is not sufficient.”

  1. Add suggestions for further research at the conclusion

Re: Suggestions for further research have been added in the conclusion section

Reviewer 2 Report

the cascade formulation of the paragraph 298-311 can be revised; not considered necessarily wrong yet for the scientific elegance of the paper it can be changed.

also, the use of "but" is considered not appropriate for a scientific paper (used a half-dozen times in the paper);  same applies for "we" as personal reference (used more than a dozen times)

Author Response

  1. the cascade formulation of the paragraph 298-311 can be revised; not considered necessarily wrong yet for the scientific elegance of the paper it can be changed.

Re: This statement has been restated, see section 3.2

“Xit represents a series of covariates that affect GTFP. including (1) R&D intensity (R&D), which is measured by the ratio of GDP to R&D internal expenditure. The larger R&D, the smaller the R&D intensity, and the R&D incentives can improve technological progress (Zhao et al., 2022), thereby increasing the city's GTFP. (2) Industrial structure (IS), which is measured by the ratio of the added value of the primary industry to the secondary industry. The larger IS, the lower the degree of industrial upgrading, and the industrial structure upgrades can reduce the proportion of polluting industries (Zheng et al., 2021). (3) Fiscal general budget expenditure (Be), which is measured by the general budget expenditure, government fiscal expenditures can encourage enterprises to carry out more green production (Li et al., 2021), thereby increasing the city's GTFP. (4) Education level (Edu), which is measured by the number of university students. The higher the overall education level of the city, the more emphasis on the concept of ecological environmental protection (Schofer et al., 2021). and it is easier to obtain human capital; (5) Transportation convenience (Tran), which is measured by the total number of passengers transported. The more convenient the city's transportation, the more convenient it is to obtain input factors such as labour force, and the higher the output factors such as GDP (Wang et al., 2021), thereby increasing the city's GTFP. (6) Internet development (Inter) which is measured by the total telecom business. The higher the level of Internet development in a city, it can improve the city's digitalization process, thereby increasing the city's productivity (Tian and Pang, 2022). (7) Electricity consumption (Ec). which is measured by the annual electricity consumption. Electricity consumption is an integral part of energy consumption, and the energy consumption is an essential factor affecting the city's GTFP (Zhang et al., 2022). All measures were expressed in logarithms except the variables R&D and IS.”

  1. also, the use of "but" is considered not appropriate for a scientific paper (used a half-dozen times in the paper); same applies for "we" as personal reference (used more than a dozen times)

Re: The "but" and "we" in the article have been modified into appropriate words, "but" is replaced by "while", "nevertheless" and "however". "we" is replaced by "this paper".

Reviewer 3 Report

Please see the attached document for detailed comments.

Author Response

  1. When the following minor corrections are made on the paper, it will become more qualified. There is an abbreviation DID in the abstract. The reader understands on page 3 that this abbreviation stands for "difference-in-difference". For those unfamiliar with this abbreviation, it would be better if the clear spelling of DID is given in the summary. The introduction section should be supported with maps and photos from Xin’an watershed, the location of 27 cities in Anhui and Zhejiang provinces for those who do not know China very well.

Re: The abbreviation of DID in the abstract has been changed to the full name, and Figure 1 has been added to mark the Xin’an watershed, the location of 27 cities in the Anhui and Zhejiang provinces.

  1. A minor typo was probably made in Figure 1. (More, Loss) should be replaced with (More, Less), (Loss, More) should be replaced with (Less, More).

Re: Thanks for the good comment. However, I don't think this part needs to be revised. Since the downstream cities have achieved ecological governance, but the upstream cities have not, the upstream city government needs to give the downstream city government ecological compensation funds, so it suffers losses instead of getting less.

  1. The paper explains in detail how the Green Total Factor Productivity (GTFP) is calculated for the Xin'an basin in Research Methodology section. However, the reader would better understand the importance and usage areas of the concept “GTFP”, if more enlightening and introductory information about why the concept of GTFP was produced and where it was used for what purpose was included in the paper.

Re: Thanks for your comment, the concept development about GTFP has been briefly added in section 3.1

“GTFP was developed based on the concept of TFP. Initially, scholars only used input factors such as labour force and capital and output factors such as GDP to measure TFP (Miller and Upadhyay, 2002; Shen et al., 2020). However, TFP lacks consideration of environmental factors. Therefore, scholars have added environmental factors as unexpected outputs to measure GTFP further so that it is more reasonable to evaluate the green eco-logical development of cities (Blicharska et al., 2022; Lin and Zhang, 2022).”

  1. In the paper, it is stated that the main obstacle for China to achieve its green ecological development and carbon neutral goal is the governance problems in watersheds located within different administrative borders:

“In 2020, it officially announced that China would strive to achieve carbon peaking by 2030 and carbon neutrality by 2060 (Lu et al., 2022). During this process, the Chinese government has paid more and more attention to the ecological governance of water ecosystems. However, there are natural obstacles to the ecological governance of water ecosystems because rivers span different administrative regions geographically, and pollution in upstream basins often causes external costs to downstream basins.” Page 2 Lines 48-53

While this statement is correct, it is an understatement because there is another important obstacle to China's achieving its green ecological development and carbon neutral goal. Recent heuristic studies (Eren, 2018; Wang et al., 2020) reveal that sustainable (green ecological) urban developments in China are largely dependent on the structure of the property market. Therefore, the issue of sustainable urban development in China should be discussed taking real estate sector structure into consideration. To increase the progressive performance of the Chinese cities towards sustainability, all-round structural improvements have to be carried out, especially with regards to the issues of property system, market transparency, property rights, illegal activities and speculation. When talking about the sustainable future of watersheds in China, it is essential to refer to the role played by the Chinese real estate market in the process:

  • Wang, X., Shi, R., & Zhou, Y. (2020). Dynamics of urban sprawl and sustainable development in China. Socio-Economic Planning Sciences, 70, 100736.
  • Eren, F. (2018) Does the Asian property market work for sustainable urban developments?. Sustainable Cities in Asia. Editors: Federico CAPROTTI, Li YU. Routledge, pp.32-47. DOI: 10.4324/9781315643069-3

Re: This has been added in the Introduction section, see section 1.

“In 2020, it officially announced that China would strive to achieve carbon peaking by 2030 and carbon neutrality by 2060 (Lu et al., 2022). During this process, the Chinese government has paid more and more attention to the ecological governance of water ecosystems. However, there are natural obstacles to the ecological governance of water ecosystems because rivers span different administrative regions geographically, and pollution in upstream basins often causes external costs to downstream basins (Dallimer and Strange, 2015). Coupling with changes in China's real estate market structure has led to more difficult green and sustainable urban development (Eren, 2017; Shen et al., 2020). Therefore, the ecological governance of rivers requires cross-governmental cooperation.”

  1. The theoretical foundation of the paper needs to be further strengthened to increase the originality and value of the work. The paper does not contain an effective theory-based literature review on watershed management. “Literature Review and Conceptual Framework” and “Conclusion” sections should be further developed. With the paper, we can understand very well China’s watershed horizontal ecological compensation policy and how this policy affects Green Total Factor Productivity (GTFP) in Anhui and Zhejiang provinces. However, we cannot see exactly how the findings of the research contribute to the theories of watershed management from a scientific point of view. In the conclusion part, findings obtained from the research should be generalized and discussed from the views of different theories of watershed management (such as ecological operation, common property theory, ecosystem services etc.). Thus, theoretical contribution of the research to the science may increase. Of course, before making this discussion, in the Literature Review and Conceptual Framework section, the views on different watershed management theories (such as ecological operation, common property theory, ecosystem services etc.) should be presented generally and intellectually, taking the ideas/subject out of the Chinese context.

Re: I am very grateful for this valuable suggestion that makes it possible to increase the contribution of this article further. I have added a statement of the relevant theory in the literature review section and summarized the relevant theory and discussion in the conclusion section. In 2.1 of the literature review section, a literature review on watershed ecological governance is added to strengthen the theoretical basis of the article.

  1. The literature section has not included some literature on theories of watershed management. I would like to ask the authors to see the papers published on this topic such as:
  • Shibao Lu, Yizi Shang, Wei Li, Xiaohe Wu, Hongbo Zhang; Basic theories and methods of watershed ecological regulation and control system. Journal of Water and Climate Change 1 June 2018; 9 (2): 293–306. doi: https://doi.org/10.2166/wcc.2018.051

  • Kerr, J. (2007). Watershed Management: Lessons from Common Property Theory. International Journal of the Commons, 1(1), 89–110. DOI: http://doi.org/10.18352/ijc.8
  • Aryal, K., Bhatta, L. D., Thapa, P. S., Ranabhat, S., Neupane, N., Joshi, J., ... & Shrestha, A. B. (2019). Payment for ecosystem services: could it be sustainable financing mechanism for watershed services in Nepal?. Green Finance, 1(3), 221-236.

Re: The above literature and recent studies have been supplemented in Section 2.1

“Cross-watershed ecological compensation is an essential tool for watershed ecological management. Watershed ecological management is closely related to maintaining ecosystem biodiversity and cyclic development (Arthington et al., 2010; Galizia Tundisi, 2018; Li and Lu, 2022). Additionally, fishery development (Jin et al., 2022), agricultural irrigation (Bruckerhoff et al., 2022) and domestic water (Cotler et al., 2022) are inseparable from this tool. However, the property rights of the same watershed belong to different ad-ministrative regions. Therefore, the success of cross-watershed ecological management requires the cooperation of governments (Li and Lu, 2022). Meanwhile, the property rights of rivers need to be clarified because the pollution of upstream watersheds is likely to cause external problems to downstream watersheds (Lu et al., 2021). Consequently, the essence of cross-watershed ecological governance is to make the upstream and down-stream watersheds common property rights among different governments and realize the ecological governance of watersheds by internalizing the external cost of pollution caused by the failure of environmental governance of upstream governments (Wang et al., 2020). Scholars in many countries have studied ecological compensation policies. Kerr (2007) predicted the difficulty of managing complex watersheds based on theories from public research, arguing that it is easier to manage at the micro-watershed level, but the governance of watersheds needs to work at the macro level, and the government needs to make trade-offs between these two approaches. Aryal et al. (2019) evaluated a pilot payment for ecosystem services (PES) program implemented in four different locations in Nepal. They found that the local government's intermediary role is critical to institutionalising the PES mechanism as a sustainable financing mechanism for securing watershed services in Nepal. Lu et al. (2018) elaborated on the importance of protecting river systems' ecological and hydrological linkages to maintain their healthy life cycles. They argued that China's lack of corresponding real-time environmental regulatory data for reservoir ecological management has led to regulatory problems. Blicharska et al. (2022) argued that ecological compensation is essential to stop biodiversity loss and natural values. They studied eco-logical compensation in Sweden and found that barriers to achieving ecological compensation were related to legislation and routines in the planning process.”

  1. The authors state that there are not many studies on the environmental regulation of watershed ecosystems:

“Although scholars have done a lot of research on environmental regulation, they mainly focus on air pollution and less research on water ecosystems.” Page 5 Lines 192-93

However, this determination does not fully reflect the truth because it is possible to find a few studies on this subject in almost every country:

  • Sulistyaningsih, T., Nurmandi, A., Salahudin, S., Roziqin, A., Kamil, M., Sihidi, I. T., & Loilatu, M. J. (2021). Public policy analysis on watershed governance in Indonesia. Sustainability, 13(12), 6615.
  • Montoya-Zumaeta, J., Rojas, E., & Wunder, S. (2019). Adding rewards to regulation: The impacts of watershed conservation on land cover and household wellbeing in Moyobamba, Peru. PloS one, 14(11), e0225367.
  • Webster, K. L., Leach, J. A., Hazlett, P. W., Fleming, R. L., Emilson, E. J., Houle, D., & Yanni, S. D. (2021). Turkey Lakes Watershed, Ontario, Canada: 40 years of interdisciplinary whole‐ecosystem research. Hydrological Processes, 35(4), e14109.

Re: Thanks for your comment and for the references that brought this issue to my attention. This section has been restated, see section 2.3

“Although scholars have done a lot of research on environmental regulation, they mainly focus on air pollution and the overall watershed. However, the heterogeneity of environmental regulation in upstream and downstream watersheds has not been fully discussed. In addition, the research on potential spatial correlations is insufficient.”

  1. The texts and numbers on the maps used in Figure 3 cannot be read easily. Higher resolution versions of these maps should be used in the paper.

Re: Complete and clear pictures have been re-uploaded

Reviewer 4 Report

The paper is very well written and contributes DID and spatial regression analysis for green total factor productivity (GTFP), which focuses on 27 cities from 2007 to 2019 in China. And the proposed scheme outperforms the state of the arts and can help policymakers pay more attention to ensuring the effectiveness of the Watershed Horizontal Ecological Compensation policy in enhancing GTFP. But, there are some problems, which must be solved before it is considered for publication. If the following problems are well-addressed, this reviewer believes that the essential contribution of this paper are important for watershed city green ecological development.

1.     For introduction, please clarify the research gaps, research questions and research objectives.

2.     For literature review and conceptual framework:

l  “Although scholars have done a lot of research on environmental regulation, they mainly focus on air pollution and less research on water ecosystems”, this sentence is not rigorous, please add previous scholars’ research and citations to support your opinion if you insist on it.

l  The methods of DEA, DID, PSM-DID, spatial model, etc. are suggested to be simply clarified in the section of literature review to clearly describe why you choose to use these methods.

3.     For research methodology, please carefully examine and clarify the formulas’ derivation process and theoretical support.

4.     For results and discussion:

l  section 4.6.2 and 4.6.5 should be added more descriptions of data to support your results.

l  I suggest revising the discussion part, which needs to be related with some previous research from literature review part.

5.     For conclusion part, the theoretical and practical implications, limitations, and future research directions should be added and clarified.

6.     Some minor revisions,

l  For the reader's convenience, when the abbreviation first appears in the paper, please fully express the name of the abbreviation. Such DID, R&D.

l  Please check the English language, grammar, and punctuation, such as Line 141, 142, etc.

l  Some texts information in the figures are missing and some figures’ resolution is low and not clear, such as Figure 1 and Figure 3.

l  Section 3.3, these data sources should be cited.

Author Response

  1. For introduction, please clarify the research gaps, research questions and research objectives.

Re: These have been revised in section 1

“After the policy was put forward, although many scholars have done research on watershed ecological governance (Sheng and Han, 2022; Wan et al., 2022), they have not discussed the heterogeneity of the policy effects of ecological governance in upstream and downstream cities, and the research on the potential spatial correlation is not sufficient. Based on this, this study uses 27 cities in Zhejiang and Anhui cities from 2007 to 2019 to analyze the direct and spatial impact of WHEC policy on green total factor productivity (GTFP) using the difference-in-differences (DID) method and carry out mechanism analysis, which is to verify whether the WHEC policy can be an effective means for China to achieve the goal of city green ecological development and carbon neutrality.”

  1. For literature review and conceptual framework:

“Although scholars have done a lot of research on environmental regulation, they mainly focus on air pollution and less research on water ecosystems”, this sentence is not rigorous, please add previous scholars’ research and citations to support your opinion if you insist on it.

Re: Indeed, this sentence is not rigorous, it has been changed, see section 2.3

"Although scholars have done a lot of research on environmental regulation, they mainly focus on air pollution and the overall watershed. However, the heterogeneity of environmental regulation in upstream and downstream watersheds has not been fully discussed. In addition, the research on potential spatial correlations is insufficient.

The methods of DEA, DID, PSM-DID, spatial model, etc. are suggested to be simply clarified in the section of literature review to clearly describe why you choose to use these methods.

Re: The reason for using each method is briefly explained in each part, for example, using DEA can more reasonably evaluate the level of green ecological development of the city, and using spatial analysis is to analyze the spatial spillover effect of policy visas, etc. Therefore, it may not be very appropriate to combine descriptions again in the literature review.

  1. For research methodology, please carefully examine and clarify the formulas’ derivation process and theoretical support.

Re: All formulas have been double-checked to make sure there are no issues.

  1. For results and discussion:

  • section 4.6.2 and 4.6.5 should be added more descriptions of data to support your results.

Re: These two parts are just to verify the robustness of the results, so there are not many descriptions, and a brief data statement supporting the results has been added. See section 4.6.2 and 4.6.5.

  • I suggest revising the discussion part, which needs to be related with some previous research from literature review part.

Re: In sections 4.3-4.6, previous studies are cited, and several recent literatures are supplemented.

  1. For conclusion part, the theoretical and practical implications, limitations, and future research directions should be added and clarified.

Re: The theoretical and practical implications, limitations, and future research directions have been added in the conclusion section

  1. Some minor revisions,

  • For the reader's convenience, when the abbreviation first appears in the paper, please fully express the name of the abbreviation. Such DID, R&D.

Re: The complete phrase has been added in the first appearance of the abbreviation in the article

  • Please check the English language, grammar, and punctuation, such as Line 141, 142, etc.

Re: Lines 141 and 142 (In the revised manuscript it is lines 183 and 184) have been revised, and the grammar and other issues of the full text have been checked and revised.

  • Some texts information in the figures are missing and some figures’ resolution is low and not clear, such as Figure 1 and Figure 3.

Re: Complete and clear pictures have been re-uploaded

  • Section 3.3, these data sources should be cited.

Re: The data sources for this section are from a different yearbook. Therefore, I think it is more appropriate to put the data source URL in the footnote.

Round 2

Reviewer 4 Report

Thanks so much for authors' revision.  I'm very appreciated to review the paper in greater depth. This is a well-written paper containing interesting results which merit publication.